# Improvement of Methods and Devices for Multi-Parameter High-Voltage Testing of Dielectric Coatings

**Vladimir Syasko [1], Alexey Musikhin [2,\*] and Igor Gnivush [3]**

[1] D.I. Mendeleev Institute for Metrology (VNIIM), Moskovsky Ave, 19, 190005 St. Petersburg, Russia; 9334343@gmail.com
[2] LLC "KONSTANTA", Ogorodny Lane, 21, 198095 St. Petersburg, Russia
[3] Faculty of Mechanical Engineering, Empress Catherine II Saint Petersburg Mining University, 21st Line, 2, 199106 St. Petersburg, Russia; klin4_g@mail.ru
[\*] Correspondence: musikhinaleksei@gmail.com; Tel.: +7-981-129-14-73

**Abstract:** Currently, the high voltage testing method is widely used to detect pinholes and porosity defects in dielectric coatings. However, most modern coatings also have requirements for the minimum allowable coating thickness. Conducting tolerance tests on the thickness of dielectric coatings concurrently along with monitoring integrity within a single technological process appears promising. Additionally, mitigating the impact of various interfering parameters is crucial. This paper conducts a theoretical and experimental examination of spark formation processes in both gas and dielectrics. This analysis takes place during the identification of both through and non-through defects in dielectric coatings on conductive substrates. The principles of selecting the test voltage for the investigated dielectric coatings, considering the need to detect both through defects and inadmissible thinning, are theoretically and experimentally justified. It is suggested to utilize a probabilistic approach for evaluating the detectability of the mentioned defects. It is demonstrated that, when the dielectric strength of the coating is known, it is feasible to identify both through and non-through defects in coatings with a calculated probability under a specified test voltage. The conditions of occurrence of partial discharges in the process of testing are investigated, and measures to suppress their influence on the inspection results are proposed. The influence of the substrate surface roughness on the magnitude of the breakdown voltage during testing is considered.

**Keywords:** dielectric strength; holiday detection; breakdown voltage; continuity; coating; thickness

## 1. Introduction

Today, testing of protective dielectric coatings is carried out in most countries on pipeline transportation infrastructure facilities, production, and other structures, and is regulated by a significant amount of regulatory documentation [1–5]. One of the main requirements for such coatings is the absence of through defects, while the need to identify non-through pores, scratches, and inadmissible thinning is not regulated, although such defects also affect the protective properties of coatings and reduce the service life of the final product [6–8].

One of the most common nondestructive testing (NDT) methods [4,5] for testing dielectric coating continuity is the high-voltage method based on the occurrence of discharges at the formation of a high-intensity electric field ($E$) between the coating surface and a conductive substrate (Figure 1) [9]. The sensitivity of the method is achieved by differences in the dielectric strength of the flawless and defective areas of the coating.

When considering the existing methods of high-voltage testing, it becomes clear that they do not pay proper attention to the connection between the coating thickness ($d_c$) and its breakdown voltage ($U_{dc}$), which is essential for the purpose of identifying specific areas of inadmissible thinning and con-through coating defects. In addition, the influence of factors caused by the parameters of the testing items is not considered (in particular, changes in the

electric field pattern at areas of roughness on the substrate surface and partial discharges caused, for example, by the undulations of the product surface), along with the conditions of inspection, in connection with which the development of a single refined methodology to detect not only con-through coating defects but also inadmissible thinning, considering the influence of interfering parameters, is a relevant task.

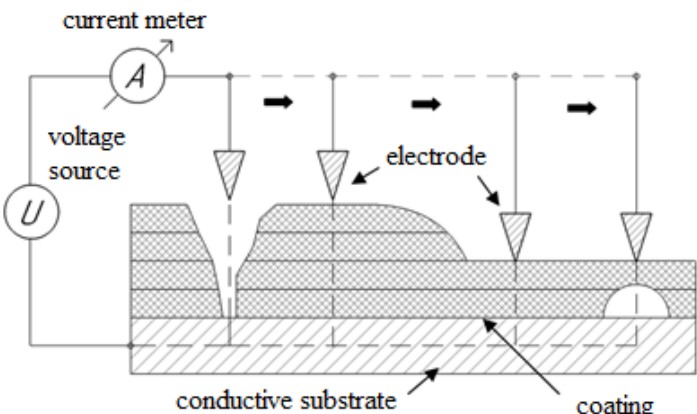

**Figure 1.** Schematic representation of the principle of the high-voltage testing method.

In [10], the authors proposed to increase the reliability of a high-voltage testing method by introducing a system of standard sample certification. However, the issues of identifying the thinning of the coating, through defects, and the degree of influence of interfering parameters on the occurrence of false positive and false negative test results were not covered in any way.

Works [11,12] describe a methodology for the integrated use of high-voltage testing methods and electric capacitance methods to identify the unacceptable thinning of cable products based on changes in the interelectrode electrical capacitance. It proposes to detect through defects according to standard methods of high-voltage testing. The works describe that thinning can be detected if its thickness is no more than 65% of the maximum cable thickness, and the final decision on the presence of thinning in the cable should be made by flaw detectors based on checking the readings with the thickness gauge readings. These results significantly increase the information content of monitoring the continuity of cable products; however, the electrical capacitance method is practically impossible to apply in the case of monitoring objects of arbitrary shape and coating thickness.

Work [6] describes an installation for automated monitoring of coating continuity, as well as a method for determining the geometric parameters of defects, such as crack size and shape. However, the influence of interfering parameters on the control process has not been studied in any way, and the testing voltage is set exclusively by standard dependencies.

To test paintwork, the electrolytic NDT method has become widespread in practice worldwide [13,14], based on the occurrence of electrical contact between the electrode and the substrate through a liquid electrolyte. This method has a number of disadvantages: the ability to control only through coating defects, the possibility of missing narrow pores due to the capillary effect, the low speed of inspection of most industrial objects, and requirements for the orientation of objects in space.

High-voltage testing of paintwork is limited due to the possibility of damaging a thin dielectric coating with high voltage, due to the small difference between the breakdown voltage of the air and the breakdown voltage of the coating (the electric hardening effect). Thus, if there is a need to test large areas of paintwork, it is necessary to pay closer attention to the choice of test voltage and reduce it to the minimum permissible (the air breakdown voltage). In addition, when performing testing using the high-voltage method, much attention should be paid to the sensitivity of the equipment, and to determining the criteria for which signals should be considered a signal about the presence of a defect in the coating. When monitoring paintwork, a situation may arise in which the discharge covers only part

of the interelectrode gap and spreads over the surface. The signal from such a discharge can be mistakenly taken for a coating defect. Accordingly, it is necessary to develop criteria for separating full and partial discharges during monitoring.

## 2. Materials and Methods

First of all, it is worth separating the tasks of detecting non-through defects and inadmissible thinning due to the difference in the physical processes of spark discharge formation in the air gap of the non-through defect and in the dielectric coating material.

The mechanism for the formation of a spark discharge for air (discharge) interelectrode gaps $d_c$ ranging from 5 μm to 50 mm is explained by the Townsend theory of electrical breakdown of gases [15,16]. If a free electron appears in a gas between two electrodes creating an electric field, then, moving towards the anode with sufficient electric field strength, it can ionize an atom or molecule of the gas upon collision. As a result, a new electron and a positive ion appear. The new electron, together with the initial one, ionizes new atoms and molecules, and the number of free electrons continuously increases until an avalanche of electrons appears. According to the above theory, a streamer is formed from electron avalanches arising in the electric field of the discharge gap, which, lengthening, covers the discharge gap and connects the electrodes, forming a spark discharge.

The intensity of electron multiplication in an avalanche is characterized by the impact ionization coefficient (the first Townsend coefficient) $\alpha$, equal to the number of ionizations produced by an electron along a path of 10 mm in the direction of action of the electric field.

When analyzing the ongoing processes, it should be taken into account that during the development of an avalanche, simultaneously with electrons, positive ions are formed, the mobility of which is much less than that of electrons, and during the development of the avalanche, they practically do not have time to move in the gap to the cathode. Thus, after the passage of an avalanche of electrons, positive ions remain in the interelectrode gap, which distorts (reduces or increases) the electric field [17].

The key element of reliable high-voltage testing is to ensure conditions for independent spark discharge in places of coating defects (for example, violation of their continuity). After the first avalanche passes through the gap, the avalanche process may resume or die out. To resume the avalanche process (organize a self-discharge), at least one secondary effective electron is required, which can arise, including as a result of the passage of a primary avalanche, with an increase in the voltage applied to the electrodes.

The number of positive ions $\left(n_i^+\right)$ remaining in the interelectrode gap after the passage of the avalanche is equal to the number of electrons in the avalanche, excluding the initial electron, i.e.,

$$n_i^+ = n_0 \cdot e^{(\alpha - \eta) \cdot d_c} - 1, \tag{1}$$

where $n_0$ is the number of primary electrons and $\eta$ is the sticking coefficient.

It should be taken into account that not all electrons knocked out from the cathode participate in the formation of secondary avalanches. Some electrons recombine with positive ions. The overall process of formation of secondary electrons from the cathode is characterized by the secondary ionization coefficient (second Townsend coefficient) $\gamma$, which depends on the cathode material, composition, and gas pressure, while $\gamma << 1$. The number of secondary electrons formed after the passage of the primary avalanche with an independent discharge form must satisfy the condition:

$$\gamma \cdot \left(e^{(\alpha - \eta) \cdot d_c} - 1\right) \geq 1 \tag{2}$$

This shows that as a result of the passage of a primary avalanche, the formation of at least one effective electron capable of igniting a secondary avalanche is necessary.

As mentioned above, during the development of an avalanche, the number of electrons and positive ions continuously increases. As the number of electrons in the avalanche head increases, the field strength at the avalanche front increases, while at the same time, the field strength decreases at the avalanche tail. This causes the electrons at the head

of the avalanche to stop and possibly recombine with ions, emitting photons, which, in turn, are able to ionize neutral molecules near the tail of the primary avalanche, forming secondary avalanches. Secondary avalanches, following the lines of force and having an excess negative charge on the head, are drawn into the region of the positive space charge left by the primary avalanche. The electrons of the secondary avalanches mix with the positive ions of the primary avalanche and form a streamer, an area with the highest current density, which, when heated, begins to glow. The highest concentration of particles (current density) is formed near the cathode. For photoionization in a gas volume, the photon energy must be greater than the ionization energy. This process is successfully carried out in mixtures of gases containing components with relatively low ionization energy (including in air).

According to the above theory [18–20], the minimum breakdown voltage of a non-through coating defect can be calculated using the following formula:

$$U_{da} = \frac{B_0 \cdot P \cdot d_c}{ln \frac{A_0 \cdot P \cdot d_c}{\ln\left(1 + \frac{1}{\gamma}\right)}}, \tag{3}$$

where $P$ is the gas pressure, $E$ is the electric field strength, $A_0$ is the coefficient that depends on the composition of the gas, $B_0$ is the coefficient which depends on the ionization energy of the gas, and $\gamma$ is the secondary electron ionization coefficient.

It can be inferred from Equation (3) that with a consistent external air temperature within a uniform field $U_{da} = f(P \cdot d_c)$, there is a quasi-constant atmospheric pressure $U_{da} = f(d_c)$ (Figure 2).

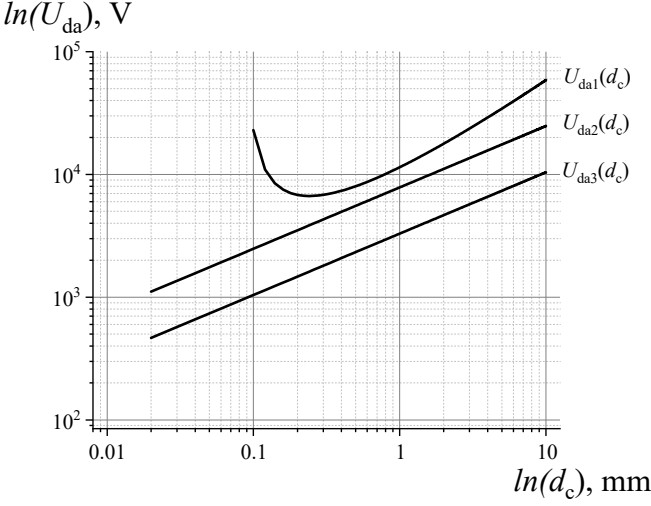

**Figure 2.** Calculated values of $U_{da}$: 1—$U_{da}(d_c)$ according to Formula (1) for atmospheric air under normal conditions, 2—$U_{da}(d_c)$ according to Formula (2) for $d_c > 1$ mm, 3—$U_{da}(d_c)$ according to Formula (2) for $d_c < 1$ mm.

Simultaneously, established techniques for determining the test voltage, accounting for the non-uniformities in the electric field during testing, provide the calculation of $U_{da}$ based on the empirical relationship [2,3]:

$$U_{da} = M \cdot \sqrt{d_c}, \tag{4}$$

where $M$ is a constant empirical coefficient depending on the thickness of the coating ($d_c$) ($M = 3294$ for coatings with $d_c < 1$ mm and $M = 7843$ for $d_c > 1$ mm).

As can be seen from Figure 2, the standardized test voltage calculation methods can be applied exclusively to detect through coating defects as they do not consider the increase in breakdown voltage of $U_{dc}$ solid dielectrics.

In [21], an empirical dependence of the coating breakdown voltage $U_{dc}$ is proposed for a wide range of dielectric coatings:

$$U_{dc} = \frac{K}{d_c} \cdot K_P \cdot \left(A_c^0\right)^{1.1} \cdot \exp\left(\frac{a}{b + lg(b)} + \frac{m}{n + lg(\tau)}\right), \quad (5)$$

where $K$ represents the proportionality factor based on $d_c$, $\tau$ signifies the duration of applied voltage, $K_P$ denotes the probability of breakdown, $A_c^0$ is the energy required for channel formation, while $a$, $b$, $n$, and $m$ are constants contingent on the dielectric for the purpose of approximation.

Equation (5) is applicable for computing the dielectric strength of dielectrics within a range of thicknesses from 0.01 to 40 mm under the duration of the applied voltage pulse $\tau = 0.1$–10 μs.

Table 1 provides a comparative display of experimentally obtained and formula-calculated $U_{dc}$ values for several solid dielectrics with a sample thickness of $d_c = 0.1$ mm [22].

**Table 1.** Calculated and experimental values of the dielectric strength $U_{dc}$ for 0.1 mm thick dielectric materials.

| Coating Material | $U_{dc}$, kV | |
|:---:|:---:|:---:|
| | Experimental Values | Calculated Values |
| polyethylene | 6.75–7 | 6.2 |
| polystyrene | 5.5–7.3 | 4.3 |
| fluoroplast-4 | 3.5 | 4 |

The larger scatter in the values $U_{dc}$ of polystyrene can be explained by its low density and, as a consequence, the large influence of the probabilistic processes of streamer formation in the material. The calculation of $U_{dc}$ using Equation (5) is applicable to both single-component and multilayer coatings, with their parameters available in the reference literature, such as those for anti-corrosion coatings on pipelines. However, it should be considered that the parameters required for the calculation of multi-component coatings (e.g., paint and varnish coatings) are usually not standardized. Therefore, $U_{dc}$ multi-component coatings or coatings for which it is not possible to determine $U_{dc}$ by calculation should be determined empirically. To experimentally determine $U_{dc}$, the recommendation is to employ a coating sample that is either identical to or closely resembles the one being tested in terms of composition and thickness, and apply it to a conductive substrate.

## 3. Results

To measure the breakdown voltages of through defects and dielectric coatings (and to calculate the electrical strengths of coatings based on the breakdown voltage), a setup was used, the structural diagram of which is shown in Figure 3. The distinctive features of each experiment are described directly in its description.

Figure 3 shows a diagram of the experimental setup. The high-voltage pulse generator creates a high voltage on the electrode (Figure 4), which is applied to the controlled sample. High-voltage pulses follow at a frequency of 50 Hz. The duration of one pulse is approximately 20 μs. Pulse amplitude can be set in the range of 0.5–40 kV. A high-voltage voltage divider is also connected to the electrode, which is needed to measure the high voltage on the electrode. The voltage divider has a division factor of 1000, and thus high voltage can be measured using a regular digital oscilloscope. When a spark discharge occurs, the discharge current flows into the discharge indication circuit. Using a current detector, the presence of a spark discharge is detected, and the device signals this using light and sound alarms.

To confirm this, an experiment was conducted to determine $U_{dc}$. Plates of textolite coated by aluminum were used as substrates, on which the paintwork was created—Molecules MLS 306 enamel (Figure 5). Enamel was utilized in three, six, and nine

layers (the thickness of one layer was equal to 12–16 μm). The ambient air temperature was controlled by a TTZh-K thermometer and varied during the experiment from 22 to 26 °C. Atmospheric pressure was controlled by the Aneroid BAMM-1 barometer and amounted to 96.2 ± 0.3 kPa.

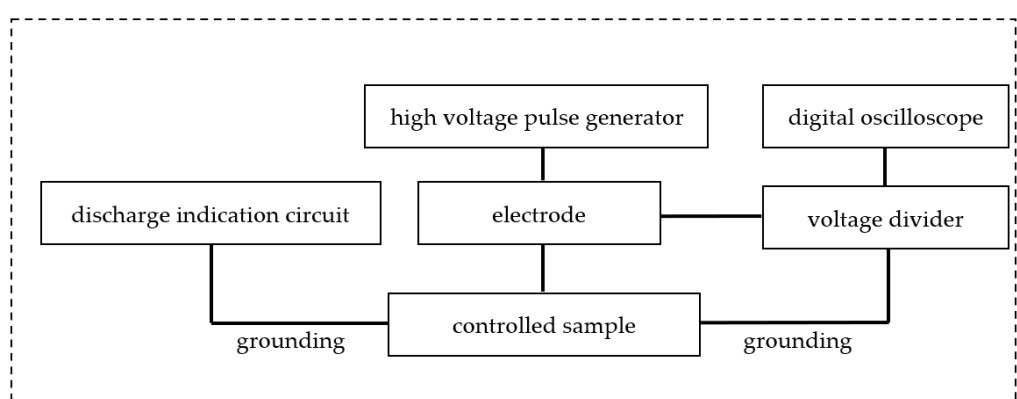

**Figure 3.** Experimental setup diagram.

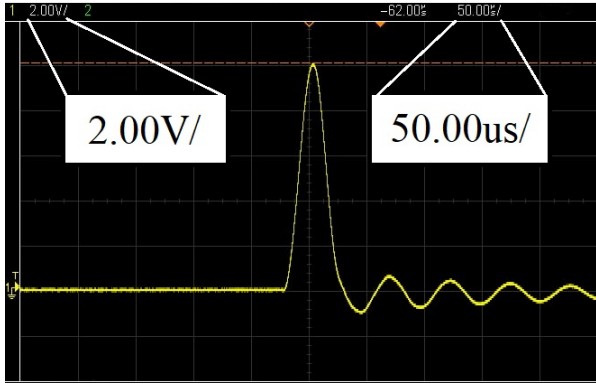

**Figure 4.** CH1: High-voltage pulse of testing voltage.

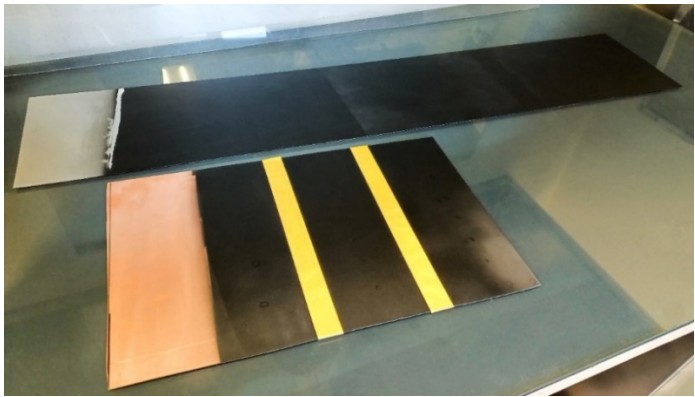

**Figure 5.** Testing objects: 1—coated aluminum sheet, 2—foil-coated laminate sheet with a fabric substrate.

After the fabrication of the specimens, the thickness was gauged at testing points where the breakdown voltage was determined. Points in this case mean the area bounded by a circle with a diameter of 5 mm. This is performed to consider the possible path of the spark discharge in the area with the lowest dielectric strength of the coating $E_c$ in this area. A test voltage was applied to the coating and increased until it broke down, and the $U_{dc}$ value was captured utilizing a DSO-X 2002A oscilloscope. Consequently, the correlation of $U_{dc}(d_c)$ was obtained, as presented in Figure 6.

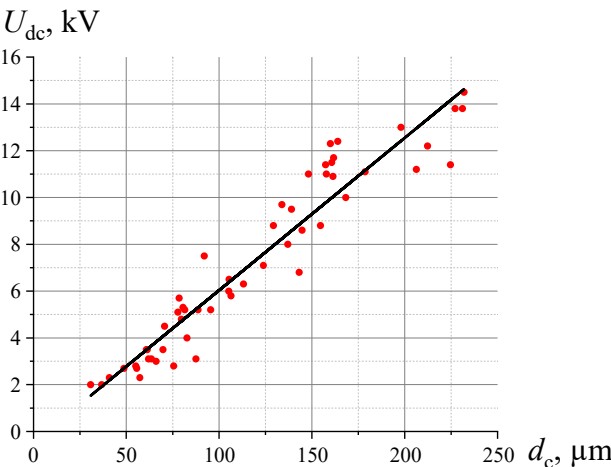

**Figure 6.** The relationship between the breakdown voltage of the coating $U_{dc}$ and its thickness $d_c$.

As evident from Figure 6, the obtained $U_{dc}$ values exhibit a considerable degree of variation. It is likely attributed to the development of discharge at the location with the least coating thickness and certain alterations in coating parameters [23,24]. Due to this factor, it is necessary to estimate the probabilities of detecting inadmissible thinning as a function of $U_i$ and $d_c$, for which an algorithm was applied to the construction of a regression line of the $U_{dc}(d_c)$ dependence and the generation of normal distribution functions with predetermined parameters from it [25–27]. In the studied area, the function $U_{dc}(d_c)$ has a quasi-linear form. Based on this, a linear regression of the type $U = k \cdot d_c + b$ was computed using the least squares method, utilizing the acquired experimental data. Subsequently, the normal distribution function of the probability of spark discharge formation from the value of $U_i$ was constructed:

$$P(U_i) = \frac{1}{\sqrt{2} \cdot \sigma} \int_{-\infty}^{U_{\text{и}}} e^{-\frac{(U_i - \mu)^2}{2 \cdot \sigma^2}} \, dU_i, \tag{6}$$

where $U_i$ is the test voltage, $k$ and $b$ are parameters of the regression line, $\mu$ is the mathematical expectation, and $\sigma$ is the standard deviation.

The boundaries of the confidence interval $P_\pm(U_i)$ concerning the regression model (Figure 7) according to [28] are as follows:

$$PP_\pm(U_i) = P(U_i) \pm t_P \sqrt{D} \sqrt{\frac{1}{n} + \frac{\left(\ln U_i - \overline{U}_i\right)^2}{\sum_{i=1}^{n}\left(\ln U_i - \overline{U}_i\right)^2}}, \tag{7}$$

where $n$ is the quantity of measurements, $t_P$ is the Student's coefficient for the 95% confidence level and $(n - 2)$ degrees of freedom, $D$ is the variance of $U_{dc}$, and $\overline{U}_i$ is the mean value of $U_i$.

The probability dependence graph of defect detection $P(U_{dc})$ indicates the dependability of the testing process. The chart, portraying a sigmoid function, delineates the boundaries of the interval at a specified confidence level (illustrated by dashed lines). Obviously, with increasing $d_c$, the characteristic $P(U_{dc})$ shifts to the right. It is also advisable to construct the relationship $P(U_{dc})$ to achieve a confidence probability of 0.9 (90%) when conducting tolerance testing for coating defects (detection of areas of inadmissible thinning) [29,30]. Therefore, it is feasible to ascertain the dielectric strength of the coating ($E_p$) with a defect detection probability set at 90% for every examined coating sample (Figure 8).

The experimental results indicate that the estimated value of $E_c$ within the designated thickness range is 75.4 ± 8.2 kV. Assuming a nearly constant dielectric strength across the specified thickness range, the likelihood of detecting a defect of a specific thickness (based on the dielectric strength of the coating) would be 0.8 (80%).

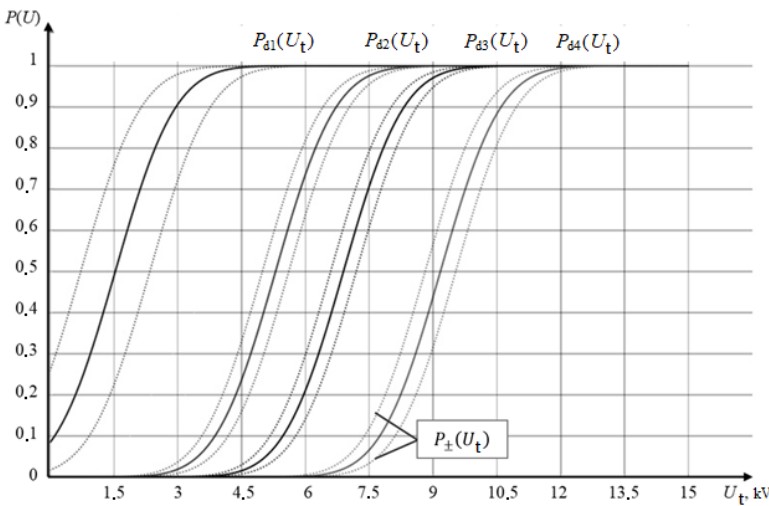

**Figure 7.** Breakdown probability distribution as a function of test voltage $U_t$ for coatings with a thickness of $d_1 = 38$ μm, $d_2 = 89$ μm, $d_3 = 113$ μm, and $d_4 = 148$ μm.

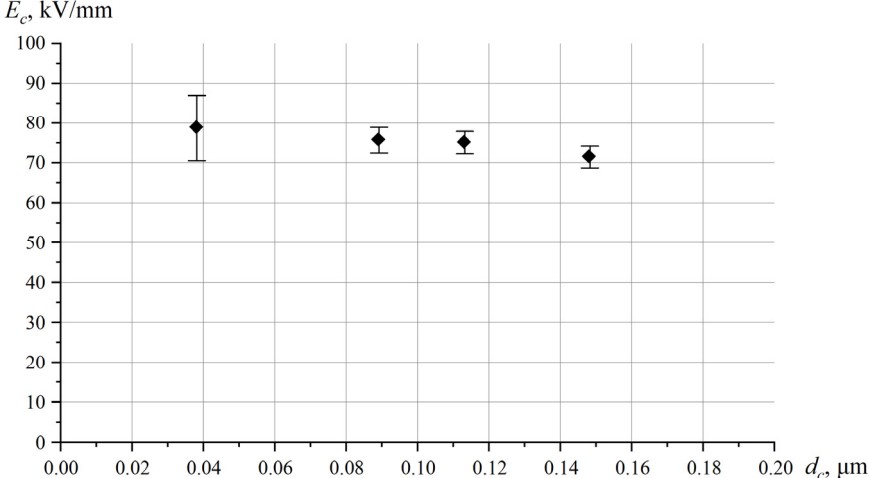

**Figure 8.** A graph showing the relationship between the dielectric strength of the coating ($E_c$) and its thickness ($d_c$) for MLS 306 coating samples.

Therefore, given a recognized dielectric strength value for the coating (determined through calculation or experimentation), one can identify unacceptable thinning with a calculated probability by conducting testing with a test voltage equal to $U_i = E_c / d_c$.

At the same time, it should be considered that during testing, including for the purpose of detecting inadmissible thinning, the formation of air gaps between the electrode and the coating surface is possible (Figure 9), which can lead to the formation of surface discharges—discharges that cover part of the interelectrode gap. Such discharges do not signal a coating defect but may generate a signal mistaken for a coating defect signal (corresponding to a full discharge) at given equipment settings. Proceeding from this, it is possible to assert that partial discharges are an interfering parameter in the process of inspection, the possibility of their occurrence should be considered, and appropriate measures should be taken to eliminate their influence when building flaw detector circuits, developing electrode designs, and creating inspection techniques.

Partial discharges occur due to air gaps ($d_a$) between the electrode and the coating, caused, for example, by inhomogeneous coating thickness, curvature or waviness of the product surface, or inaccuracy in placing the electrode on the coating [31]. In the formed air gaps, the electric field strength can exceed the field strength in the dielectric coating because the relative dielectric constant of the coating is greater than the dielectric constant of the air

(Figure 10). In this case, favorable conditions for the formation of partial discharges are created [11,32].

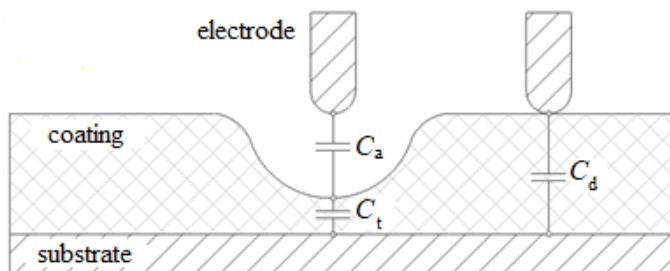

**Figure 9.** Equivalent scheme to explain the processes of partial discharge occurrences in the testing area; $C_a$ is the air gap capacitance; $C_d$ is the dielectric capacitance in the area of inadmissible thinning (coating defect); and $C_a$ is the dielectric capacitance in the coating area of the specified thickness.

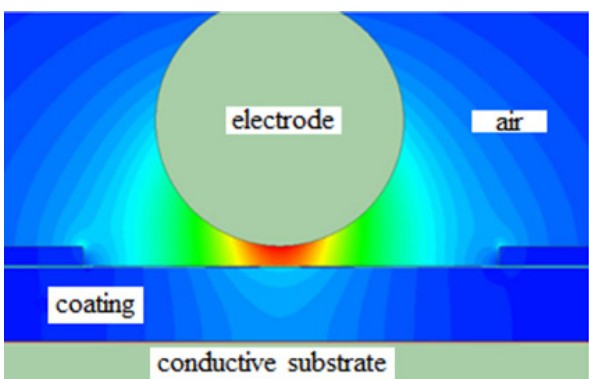

**Figure 10.** Finite element model of electric field strength distribution on a section with air gap 50 μm and residual coating thickness 180 μm.

The primary informative parameter carrying information about the presence or absence of defects in the coating is the amplitude of the voltage pulse on the shunt $u_R(t)$, included in series in the measurement circuit when the discharge current flows through it. It has been experimentally established [33] that in the area of small coating thickness ($d_c \approx 50$ μm) $u_R(t)$ at full discharge, the current flowing through it is smaller than the amplitude of $u_R(t)$ at partial discharge in the case of large coating thickness (Figure 11). It follows that for instrumental realizations of the nondestructive electric discharge testing, it is necessary to adjust the sensitivity of the instruments. At the same time, if the sensitivity of the device is set incorrectly, false alarms due to partial discharges may occur.

Adjustment of the required sensitivity level can be carried out on test or standard samples identical to or close in characteristics to the testing object. However, this approach does not exclude the possibility of human factor influence on the choice of the sensitivity level and, as a consequence, on the testing results.

Figure 12a,b shows the oscillograms of voltages at the formation of test voltage pulses and the occurrence of partial discharges: pulse $u_i(t)$ of the test voltage (in blue) and pulses of voltage drop on the shunt $u_R(t)$, included in series in the measurement circuit, are caused by the flow of partial discharge currents (in red). Figure 12c, shows the oscillograms of $u_i(t)$ and $u_R(t)$ pulses at full discharge through the interelectrode gap in the area of the through coating defect.

As can be seen from the oscillograms, in general, the amplitude of $u_R(t)$ at partial discharge is smaller than the amplitude of $u_R(t)$ at full discharge. This is due to the fact that in a partial discharge, the source of charges is the charged capacitance of the air gap ($C_a$), while in a full discharge, the current is due to the flow of charges from both the charged capacitances and the high voltage source. As a consequence, at the moment of complete

discharge of the interelectrode gap, the voltage at the electrode $u_i(t)$ decreases to close to zero and, accordingly, the duration $\tau$ of the test voltage pulse decreases. Thus, in order to completely eliminate false positives due to partial discharges, it is proposed to estimate $\tau$ for $u_i(t)$ and fix the decrease in $\tau$ at full discharge using the scheme, the structure of which is presented in Figure 11.

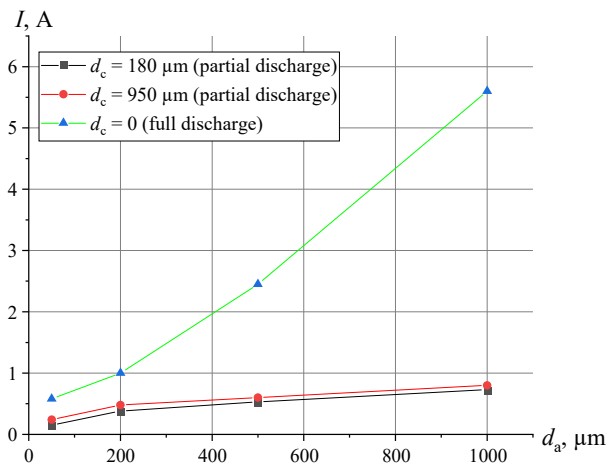

**Figure 11.** Dependence of the amplitude $I$ of the discharge current pulse and calculated values of the partial discharge current on the size of the air gap $d_c$ for three values of the coating thickness $d_c$.

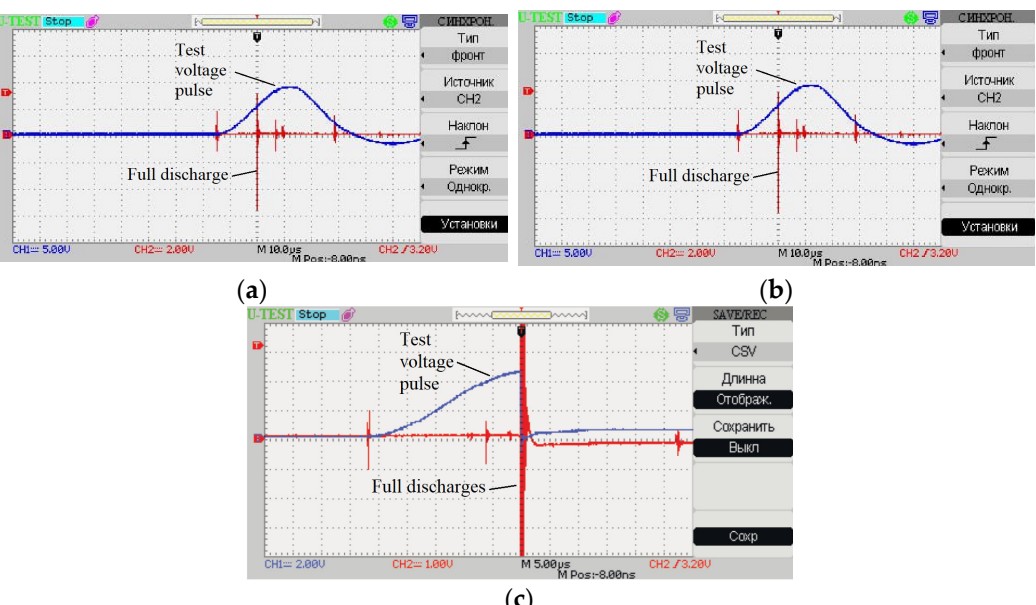

**Figure 12.** Oscillograms of test voltage pulses and partial discharges: (**a**) without full discharge (time sweep 10 µs/div., scale: CH1: 5 V/div., CH2: 2 V/div.); (**b**) without full discharge (time sweep 2.5 µs/div., scale: CH1: 2 V/div., CH2: 1 V/div.). (**c**) at full discharge (time sweep 5 µs/div., scale: CH1: 2 V/div., CH2: 1 V/div.).

The circuit proposed in Figure 13 works as follows: the voltage $u_R(t)$, the amplitude of which depends on the value of the discharge current, is fed through a low-pass filter to the comparator driver, which limits the maximum amplitude of the pulse. The comparator compares the pulse amplitude $u_R(t)$ with the set sensitivity level, and the output is a logic one or logic zero. At the same time, the circuit measures the duration $\tau$ of the pulse $u_i(t)$ of the test voltage whose amplitude is divided by a factor of 1000. The divided pulse is fed through a rectifier to cut off the negative component of the pulse. The rectified pulse

is fed to a comparator, which converts the analog pulse into a digital rectangular pulse (meander). The duration $\tau$ of this pulse is measured by the timer of the microcontroller.

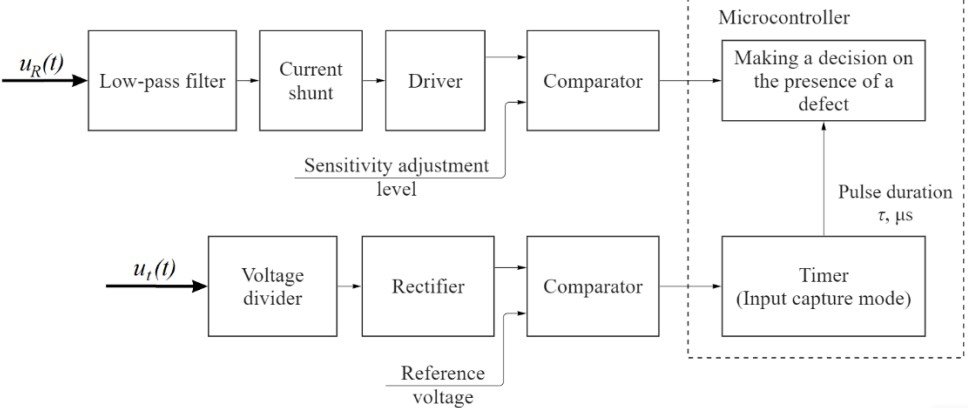

**Figure 13.** Structural diagram of the defect registration unit.

The scheme proposed in Figure 13 realizes the tracking of the coincidence of two conditions in the defect area: exceeding the sensitivity threshold by the voltage pulse $u_R(t)$ caused by the full discharge current and reducing the duration of the pulse $u_i(t)$, which eliminates the possibility of false positive testing results caused by partial discharges.

However, false positive testing results can also occur due to the increased roughness of the substrate. In [34,35], the influence of the inhomogeneity of the electric field formed in the testing region on the breakdown voltage of the method was shown. In these works, it is proposed to form a sharply inhomogeneous field to reduce the breakdown voltage, but the influence of the roughness of the substrate surface on the inhomogeneity of the field was not considered.

It is known that surface roughness is characterized, as a rule, by the arithmetic mean deviation of the profile along the substrate length ($R_a$) and the height of profile irregularities along ten points ($R_z$) [36]. Surface roughness parameters are set in accordance with [37]. In electric discharge testing, the substrate is one of the electrodes, the shape of which also determines the pattern of the electric field in the interelectrode gap. In this case, in a highly inhomogeneous field, regions of increased electric field strength appear, resulting in a decrease in the breakdown voltage of the gas gap. Thus, one can talk about the possible influence of surface roughness on the breakdown voltage. If the roughness is significant, the substrate should not be considered as a plane in the system of two electrodes, but as a sequence of irregularities with protrusions and depressions (Figure 14), leading to an increase in the degree of inhomogeneity of the electric field, which may entail a decrease in the value of the breakdown voltage.

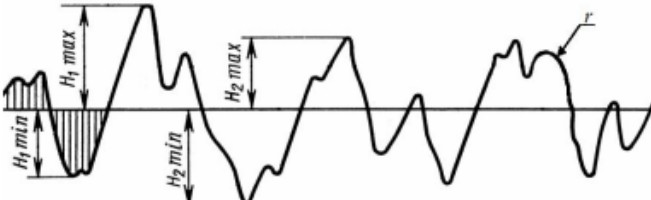

**Figure 14.** Surface roughness and parameters generally characterizing it.

To evaluate the effect of substrate surface roughness (e.g., after sand or shot blasting) on breakdown voltage, experiments were conducted with an Elcometer 125 Surface Comparator on shot roughness samples. In the first case, a 0.05 mm thick film with a hole simulating a defect was mounted on the sample. An electrode was placed on the surface of the film in the area of the hole. The structure was fixed with clamps (Figure 15a). In the second case, the roughness samples were painted with MLS 306 enamel (Figure 15b). In

both cases, the spark breakdown occurrence voltage was recorded for different values of surface roughness.

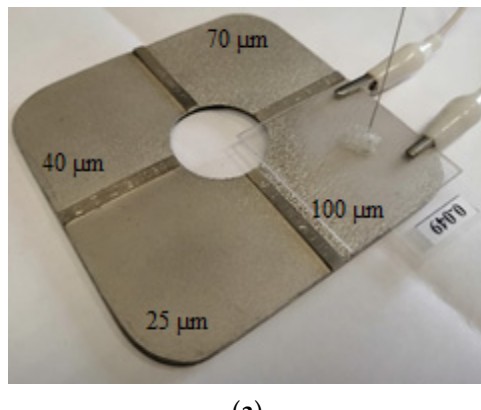

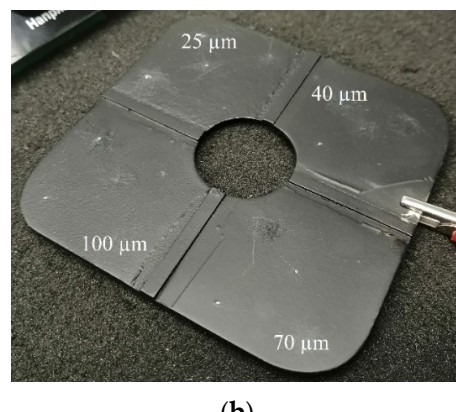

(**a**)                                                                                                       (**b**)

**Figure 15.** Schematic diagram of the experiment to investigate the influence of roughness. The experimental results are summarized in Tables 2 and 3.

**Table 2.** Results of air gap breakdown voltage measurements for different values of $R_z$ of the substrate in the film experiment.

| $R_z$, μm | $U_{dc}$, kV | | |
|:---:|:---:|:---:|:---:|
| 25 | 1.3 | 1.2 | 1.3 |
| 40 | 1.3 | 1.3 | 1.26 |
| 70 | 1.36 | 1.3 | 1.28 |
| 100 | 1.36 | 1.3 | 1.36 |

**Table 3.** Results of air gap breakdown voltage measurements for different values of $R_z$ of the substrate in the enamel experiment.

| $R_z$, μm | $U_{dc}$, kV | | |
|:---:|:---:|:---:|:---:|
| 25 | 0.85 | 0.80 | 0.85 |
| 40 | 0.85 | 0.85 | 0.80 |
| 70 | 0.85 | 0.90 | 0.85 |
| 100 | 0.85 | 0.90 | 0.90 |

It can be seen from the data obtained that the breakdown voltage of the interelectrode gap simulating the through coating defect is practically unchanged when $R_z$ is changed in the range of up to 100 μm. Presumably, this is due to the fact that the diameter $D$ of the through defect is much larger than the distance $T$ between neighboring protrusions, hence the spark discharge always occurs in the region between the peak and the rod. Thus, one can conclude that for defects with a diameter $D \gg T$, the roughness of the substrate has no significant influence. On the other hand, if $D < T$, the defect may form in the trough region, leading to an increase in the interelectrode gap and an increase in the breakdown voltage. In such a case, it is proposed to calculate the test stress substrated on the value $d_{cm} + \frac{1}{2}R_z$, where $d_{cm}$ is the maximum thickness of the testing coating.

## 4. Discussion

The theoretical analysis of spark formation mechanisms in gas and solid bodies, as well as the use of the regression algorithm for processing experimental results, made it possible to develop the main provisions of the methodology for identifying areas of inadmissible

thinning in dielectric anti-corrosion coatings. High-voltage testing using the proposed technique increases informativeness while ensuring highly reliable results.

At the same time, the probabilistic nature of the occurrence of breakdown, the heterogeneity of the thickness of the coating samples, and the possible presence of defects in paint coating samples that locally worsen the electrical strength of coating samples contributed to an increase in the spread of experimental values of breakdown voltage, which, at this stage, makes it possible to identify, using the electric spark method, thinning only when the residual coating thickness is approximately 50% of the nominal thickness or less (this may be due to an inappropriate number of paint layers). To reduce the scatter, it is necessary to conduct further studies on interfering factors and methods for the local measurement of coating thickness under conditions of increased external electric field strength.

On the other hand, experimental and theoretical studies on the influence of partial discharges made it possible to formulate an algorithm for signal processing in electric discharge testing and a structural scheme of the device, which considers an additional informative parameter for making a decision about the presence of a defect in the coating and provides tuning from the influence of partial discharges on the inspection result. However, at this stage, tests of the instrumental implementation of the algorithm shown in Figure 11 were carried out only for samples of paintwork and organic glass. In the future, it is planned to conduct tests on a larger number of control objects (the external and internal coatings of pipes, roofing coatings, industrial paint, and varnish coatings).

Finally, theoretical and experimental analysis of the effect of the substrate roughness on the breakdown voltage of the interelectrode spacing of a given thickness has shown that there is no significant effect of the substrate surface roughness on the breakdown voltage of the interelectrode spacing in the range of $R_z$ from 25 to 100 μm.

The obtained results make it possible to use the high-voltage method of non-destructive testing to identify through defects and an unacceptably small number of layers of paint and varnish coatings, which was previously inaccessible for testing by high-performance methods and was detected mainly visually.

**Author Contributions:** Conceptualization, V.S. and A.M.; methodology, V.S.; software, I.G.; validation, V.S., A.M. and I.G.; formal analysis, V.S., A.M. and I.G.; investigation, V.S., A.M. and I.G.; resources, V.S.; data curation, V.S., A.M. and I.G.; writing—original draft preparation, V.S., A.M. and I.G.; writing—review and editing, V.S., A.M. and I.G.; visualization, A.M. and I.G.; supervision, V.S., A.M. and I.G.; project administration, V.S. All authors have read and agreed to the published version of the manuscript.

**Funding:** This research received no external funding.

**Institutional Review Board Statement:** Not applicable.

**Informed Consent Statement:** Not applicable.

**Data Availability Statement:** The original contributions presented in the study are included in the article, further inquiries can be directed to the corresponding author/s.

**Conflicts of Interest:** Author Alexey Musikhin was employed by the company LLC "KONSTANTA", Ogorodny Lane, 21, 198095 Saint Petersburg, Russia. The remaining authors declare that the research was conducted in the absence of any commercial or financial relationships that could be construed as a potential conflict of interest.

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
