# Peer review of "Improvement of Methods and Devices for Multi-Parameter High-Voltage Testing of Dielectric Coatings"

_coatings, doi:10.3390/coatings14040427_

Round 1

Reviewer 1 Report (Previous Reviewer 1)

Comments and Suggestions for Authors

The article is now well organized and clear, I recommend publishing it.
It is necessary to check the typos in the text, I noticed some minor mistakes:

line 26 - Keywords: keyword 1; diel ........

line  38 - ...high-intensity electric field E between the...  Electric field in italic with commas or delete

line 44/45 - similarly

...relationship between the thickness dc of coatings and their 44
breakdown voltage Udc for the task of identifying areas of inadmiss.....

Regatds

Comments on the Quality of English Language

fine

Author Response

The team of authors expresses deep gratitude to the reviewer for the work done.

We agree with the comments regarding typos. Additional editing of the article was carried out to look for additional typos.

Reviewer 2 Report (New Reviewer)

Comments and Suggestions for Authors

1. The literature review (Introduction) is performed incorrectly. There are too few references to literature. There is no systematization of knowledge. The authors did not prove a research gap based on the literature review.

2. A description of the methods used and a description of the experiment are clearly missing.

3. There is no discussion about the results, possible errors, and the causes of errors.

4. There is a lack of discussion of the results (key insights) that would be significant for engineers in the field.

Comments on the Quality of English Language

Extensive editing of English language required.

Author Response

The team of authors thanks the reviewer for the work done.

There are the responses to the comments point by point.

1.       The literature review (Introduction) is performed incorrectly. There are too few references to literature. There is no systematization of knowledge. The authors did not prove a research gap based on the literature review.

Response: In the Introduction section, lines 52 to 85, a review of related studies and argumentation of the goals and objectives of the current study have been added.

2.       A description of the methods used and a description of the experiment are clearly missing.

Response: Before each described experiment, a description of tested object and the equipment used is given. A detailed description of the experimental setup has been added to determine the breakdown voltage of air gaps and dielectric coatings during experiments (lines 187-213).

3.       There is no discussion about the results, possible errors, and the causes of errors.

Response: In the discussion section (lines 390...) they described the results obtained, the positive effect of the results obtained, as well as possible errors and inaccuracies of the study.

4.     There is a lack of discussion of the results (key insights) that would be significant for engineers in the field.

Response: The key benefits of the results are discussed in the text after each parameter has been examined and are also summarized in the discussion section.

Reviewer 3 Report (New Reviewer)

Comments and Suggestions for Authors

Review

Manuscript ID: coatings-2890096

Comments and Suggestions

The research article titled "Improvement of methods and devices of multi-parameter high voltage testing of dielectric coatings" presents a thorough investigation into enhancing high voltage testing methodologies for detecting pinholes, porosity defects, and inadmissible thinning in dielectric coatings. The research addresses an important issue in defectoscopy, focusing on the critical task of identifying not only through defects but also non-through defects like inadmissible thinning in dielectric coatings. This is crucial for ensuring the longevity and effectiveness of protective coatings. The theoretical and experimental analyses conducted to understand spark formation processes in gas and dielectrics are commendable. These analyses provide a solid foundation for the proposed methodologies. The proposed probabilistic approach for assessing the detectability of defects based on known dielectric strength is innovative and could significantly enhance testing reliability. The discussion on the influence of substrate surface roughness on breakdown voltage provides valuable insights for refining testing procedures, contributing to the advancement of the field.

Suggestions for Major Revision:

1. Provide a clearer delineation of the research objectives and hypotheses at the outset of the article to guide readers through the study's aims and scope.

2. Enhance the organization and flow of the introduction section by structuring it to lead logically into the subsequent sections, providing a smoother transition for readers.

3. Strengthen the theoretical underpinnings by providing more detailed explanations of the Townsend discharge theory and other relevant theoretical frameworks.

4. Clearly define the experimental procedures, including sample preparation, testing protocols, and data collection methods, to ensure reproducibility and transparency.

5. Improve clarity and precision in the presentation of results, including figures and tables, to facilitate easier interpretation by readers.

6. Expand the discussion section to provide deeper insights into the implications of the findings and their relevance to practical applications in defect detection and coating evaluation.

7. Incorporate a more explicit discussion on the limitations and potential sources of error in the proposed methodologies, along with suggestions for addressing these challenges in future research.

8. Consider integrating theoretical deductions or models to further elucidate the observed experimental phenomena and enhance the theoretical framework of the study.

Recommendation for Review Decision:

Based on the comprehensive analysis presented and the potential significance of the proposed methodologies for improving high voltage testing of dielectric coatings, I recommend accepting the article for publication in Coatings after the major revisions as suggested above. The thorough investigation, innovative methodologies, and empirical evidence presented in the study contribute valuable insights to the field of defectoscopy and coating evaluation.

Comments on the Quality of English Language

Extensive editing of English language required.

Author Response

The team of authors expresses their deep gratitude to the reviewer for the work done and recommendations for improving the article.

There are the responses to the comments point by point below.

1. Provide a clearer delineation of the research objectives and hypotheses at the outset of the article to guide readers through the study's aims and scope.

Response: We agree with the comment and in the Introduction section, lines 52 to 85, a review of related studies and argumentation of the goals and objectives of the current study have been added.

2. Enhance the organization and flow of the introduction section by structuring it to lead logically into the subsequent sections, providing a smoother transition for readers.

Response: At the end of each section, semantic transitions were added to make the material easier to understand for the reader.

3. Strengthen the theoretical underpinnings by providing more detailed explanations of the Townsend discharge theory and other relevant theoretical frameworks.

Response: In the Materials and Methods section (86 – 141 lines) we added a detailed explanation of the mechanisms of discharge formation in gases according to Townsend.

4. Clearly define the experimental procedures, including sample preparation, testing protocols, and data collection methods, to ensure reproducibility and transparency.
Response: From line 186, a description of the experimental setup was added to determine the high voltage parameters for various experiments. For each experiment, additional standardized parameters of the environment, the tested object and the equipment used are listed.

5. Improve clarity and precision in the presentation of results, including figures and tables, to facilitate easier interpretation by readers.
Response: According to the authors, the graphic materials sufficiently reflect the results of the study. If, in the opinion of the reviewer, the work contains data/results that are incomprehensible, please indicate more specifically.

6. Expand the discussion section to provide deeper insights into the implications of the findings and their relevance to practical applications in defect detection and coating evaluation.

Response: We agree with the comment and in the Discussion section (lines 390....) we described the results obtained, the positive effect of the results obtained, as well as possible errors and inaccuracies of the study.

7. Incorporate a more explicit discussion on the limitations and potential sources of error in the proposed methodologies, along with suggestions for addressing these challenges in future research.

Response: We agree with the comment and Possible errors were added to Discussion section.

8. Consider integrating theoretical deductions or models to further elucidate the observed experimental phenomena and enhance the theoretical framework of the study.

Response: In the Discussion a discussion of future research prospects was included.

Round 2

Reviewer 2 Report (New Reviewer)

Comments and Suggestions for Authors

All of my questions have been answered. Thank you.

Comments on the Quality of English Language

Moderate editing of English language required.

Reviewer 3 Report (New Reviewer)

Comments and Suggestions for Authors

The authors of the paper have made thorough and careful revisions in response to the comments raised by the reviewers. Therefore, I recommend accepting this paper for publication.

Comments on the Quality of English Language

Extensive editing of English language required.

This manuscript is a resubmission of an earlier submission. The following is a list of the peer review reports and author responses from that submission.

Round 1

Reviewer 1 Report

Comments and Suggestions for Authors

Methods of predicting electrical breakdown (especially in finished products) are of great importance. This article touched on that topic. Since this article is also intended for the scientific community, more details are needed. I suggest that this article be considered after the general revision.

The main objections:

- Bad language, unusual terms, such as "The technique of holiday..." should be removed from the article. Too long sentences 12-14 "It seems promising to carry out testing of the dielectric coatings thickness in one technological process with testing of their continuity, by changing existing methods of testing and a method of forming the test voltage". The final version of the article should be written more carefully.

- The affiliations should be completed with the name of the country.

- The biggest complaint is poor citation and comparison of presented results with similar ones by other authors.

- In the presented equations, it is not clear what the author's contribution is. In line 67, the serial number of the equation is missing.

- More details about the experimental conditions are necessary, it is not enough to mention the standards.

The article can be evaluated only after the correction of the mentioned remarks.

Reviewer 2 Report

Comments and Suggestions for Authors

Thank you very much. You showed some figures for paper. But, there are some comments.

1.       The number of reference is low. Author must be described about some literature introduction.

2.       Author must be observed about rough on surface of material. And, author must be added about data of roughness of surface of material.

3.       Author showed about breakdown voltage and coating thickness in figure 3 (p5). The scattering of breakdown voltage became big when coating thickness was thick.

4.       Author must be added about some values of temperature on some surface areas of material.

5.       Author showed about ln-ln curves in figure 2. Author should be changed about axis of curve.(figure 2)

6.       Please clear about channeling energy, a,b. n, m – some constants thickness of 0.01 to 40 mm with an applied voltage pulse duration τ= 0.1-10um.(p4)

7.       The thickness of one layer is not clear.(p4)

8.       The scattering of Polystyrene became big. Author should be described the phenomena (Table 1).

9.       The coating material in layer is not clear in paper.

Reviewer 3 Report

Comments and Suggestions for Authors

This article presents the technique of holiday and unacceptable thinning detection 2 in dielectric coatings by high voltage testing. This manuscript can be accepted after following revisions.

1.       Authors looks premature while writing manuscript. In introduction sections there are many paragraphs. Please rewrite introduction section.

2.       The formula equation should be cited.